# PRMT5 Identified as a Viable Target for Combination Therapy in Preclinical Models of Pancreatic Cancer

**DOI:** 10.3390/biom15070948

**Published:** 2025-06-30

**Authors:** Xiaolong Wei, William J. Kane, Sara J. Adair, Sarbajeet Nagdas, Denis Liu, Todd W. Bauer

**Affiliations:** 1Department of Surgery, University of Virginia School of Medicine, Charlottesville, VA 22908, USA; xw8h@virginia.edu (X.W.); wjk2a@uvahealth.org (W.J.K.); sjs7w@uvahealth.org (S.J.A.); dl6uv@virginia.edu (D.L.); 2Department of Microbiology, Immunology, and Cancer Biology, University of Virginia School of Medicine, Charlottesville, VA 22908, USA; sn9mn@virginia.edu

**Keywords:** pancreatic cancer, PRMT5, therapeutic target, gemcitabine, paclitaxel, DNA damage, combination therapy, orthotopic PDX mouse model

## Abstract

Pancreatic cancer is the third leading cause of cancer-related death in the US. First-line chemotherapy regimens for pancreatic ductal adenocarcinoma (PDAC) include FOLFIRINOX or gemcitabine (Gem) with or without paclitaxel (Ptx); however, 5-year survival with these regimens remains poor. Previous work has demonstrated protein arginine methyltransferase 5 (PRMT5) to be a promising therapeutic target in combination with Gem for the treatment of PDAC; however, these findings have yet to be confirmed in relevant preclinical models of PDAC. To test the possibility of PRMT5 as a viable therapeutic target, clinically relevant orthotopic and metastatic patient-derived xenograft (PDX) mouse models of PDAC growth were utilized to evaluate the effect of PRMT5 knockout (KO) or pharmacologic inhibition on treatment with Gem alone or Gem with Ptx. Primary endpoints included tumor volume, tumor weight, or metastatic tumor burden as appropriate. The results showed that Gem-treated PRMT5 KO tumors exhibited decreased growth and were smaller in size compared to Gem-treated wild-type (WT) tumors. Similarly, the Gem-treated PRMT5 KO metastatic burden was lower than the Gem-treated WT metastatic burden. The addition of a PRMT5 pharmacologic inhibitor to Gem and Ptx therapy resulted in a lower final tumor weight and fewer metastatic tumors. The depletion of PRMT5 results in increased DNA damage in response to Gem and Ptx treatment. Thus, PRMT5 genetic depletion or inhibition in combination with Gem-based therapy improved the response in primary and metastatic PDAC in clinically relevant mouse models, suggesting that PRMT5 is a viable therapeutic target for combination therapy in PDAC.

## 1. Introduction

Pancreatic cancer is the third leading cause of cancer-related death in the US and is projected to be the second leading cause of cancer-related death by 2030 [1,2]. Although it has slowly improved, the five-year survival rate for all patients with pancreatic cancer remains poor, around 9~13% [3]. For pancreatic ductal adenocarcinoma (PDAC), first-line chemotherapy regimens include combination treatment with fluorouracil, irinotecan, oxaliplatin, and leucovorin (FOLFIRINOX) or gemcitabine (Gem) with or without paclitaxel (Ptx) [4]. Given the severe toxicities associated with FOLFIRINOX, Gem with or without Ptx is preferred in many patients with advanced or metastatic disease. Despite advances in these therapies, survival in patients with unresectable or metastatic PDAC has remained largely unchanged over the last 50 years [5]. As such, new and effective therapies are desperately needed [6].

Our group previously performed targeted in vivo CRISPR/Cas9 genetic screening, which led to the identification of the protein arginine methyltransferase 5 (PRMT5) as a possible therapeutic target against pancreatic cancer when combined with Gem [7,8]. *PRMT5* is an oncogene involved in protein arginine methylation that is associated with tumor cell proliferation, differentiation, invasion, and migration [9,10,11]. Moreover, PRMT5 is overexpressed in many human cancers and has been increasingly recognized as a promising anticancer target [12,13,14,15,16,17]. Accordingly, several different PRMT5 inhibitors have being investigated in phase I clinical trials for solid organ malignancies [18,19,20,21,22,23]. However, a new combination therapy with these inhibitors or PRMT5 depletion with chemotherapeutics will need to be tested in preclinical models.

Importantly, in validating our group’s discovery of PRMT5 as a potentially lethal target in combination with Gem, a subcutaneous mouse model of PDAC was employed. However, previous work has emphasized that such heterotopic models offer an insufficient representation of the PDAC microenvironment, thus making them poor predictors of the human response to anticancer therapy [24]. Similarly, the use of a commercial cell line in this model restricted the generalizability of our initial findings. Given the limitations of these prior models and the increasing scientific and clinical interest in PRMT5, we sought to confirm the efficacy of PRMT5 as a target in PDAC with clinically relevant orthotopic and metastatic patient-derived xenograft (PDX) mouse models. If successful, such results would pave the way for a clinical trial of PRMT5 in combination with Gem in patients with PDAC. In addition, we sought to explore possible therapeutic escape mechanisms and identify candidate targets that may enhance the anticancer effect of PRMT5. Thus, this study aims to investigate the possibility of a new combination therapy involving PRMT5 depletion or inhibitors with chemotherapeutics using PDX models. These findings will provide novel insights into developing new combination therapies for patients with PDAC.

## 2. Materials and Methods

### 2.1. Cell Lines and Patient-Derived Xenografts

Genetic knockout (KO) of *PRMT5* in mPanc96 PDAC cells was performed with CRISPR/Cas9 and validated as previously described [8]. Wild-type (WT) and *PRMT5* KO mPanc96 cells were transduced using firefly luciferase lentivirus (KeraFAST, Boston, MA, USA) to allow for metastatic model imaging as described below.

Tumor 366, a *KRAS*-mutant PDX originating from a human PDAC specimen that was collected, examined, and propagated within immunocompromised mice as previously described [25,26,27], was used in this study. All protocols were performed with the approval of the Institutional Review Board of the University of Virginia in coordination with the Biorepository and Tissue Research Facility. All patients provided written consent for participation. This study was carried out in strict accordance with the recommendations in the Guide for the Care and Use of Laboratory Animals by the National Institutes of Health [28]. The protocol was approved by the Animal Care and Use Committee of the University of Virginia (Approval code 4078, 12 May 2020).

### 2.2. PRMT5 KO in the Orthotopic Pancreatic Injection (P.I.) Model

A total of 1 × 10^6^ wild-type (WT) or *PRMT5* KO mPanc96 cells suspended in 50 uL of media were orthotopically injected into the pancreas of 6-to-8-week-old male athymic nude mice (Envigo, Indianapolis, IN, USA) utilizing the technique previously described [7,8]. Briefly, mice were anesthetized with 0.1 cc of ketamine (75 mg/kg intraperitoneal injection [I.P.]) and dexmedetomidine (0.2 mg/kg I.P.), their left flank was incised and opened, and the pancreas was exteriorized. Following cell injection and confirmation of a pancreatic subcapsular bubble, the pancreas was returned into the abdomen and the flank was closed in two layers, followed by anesthesia reversal (atipamezole 2 mg/kg subcutaneous injection [S.Q.]) and analgesia (ketoprofen 4 mg/kg S.Q.). On post-operative day 4, mice were randomized to groups with no therapy or treatment with Gem (100 mg/kg I.P. twice weekly). A total of 40 mice (10 mice in each group) were used in this study. Tumor growth was monitored on a weekly basis with volumetric magnetic resonance imaging (MRI) starting at 3 weeks post-operatively, when tumors had achieved a volume of 100–500 mm^3^. Mice were observed throughout all experiments and euthanized when tumors reached a volume prohibitive to pain-free survival, developed ascites, experienced 20% weight loss, or met humane endpoints. Mice were euthanized via isoflurane anesthesia, followed by cervical dislocation, and tumor tissues were collected for weighing and further analysis.

MRI was carried out with a 7.0T ClinScan MRI system (Siemens/Bruker; Siemens Corp, Munich, Germany) as previously described [25,26]. Briefly, after mice were anesthetized, MRI data were collected by generating 0.5 mm axial imaging slices, covering the entire tumor. PACS (Kodak Carestream, Rochester, NY, USA) was used to measure the tumor area for each image slice. Slice volume was calculated by multiplying the slice thickness (0.5 mm) by the sum of the areas of tumor visible (mm^2^) in each slice. Tumor volume was calculated using the following formula: Tumor volume = (Slice volume 1 + Slice volume 2 + Slice volume 3…).

At humane endpoints, mice were also autopsied to assess liver or abdominal metastasis. For each mouse, if metastasis was found, they were scored 1 point, and if no metastasis was found, they were scored 0 points. In each group, the percentage of metastasis (%) = sum of scores ÷ mice number, or the percentage of mice in the group with metastasis.

### 2.3. PRMT5 KO in the Metastatic Splenic Injection (S.I.) Model

A total of 2 × 10^6^ WT or *PRMT5* KO luciferase-transduced mPanc96 cells suspended in 50 uL of serum-free media were injected into the spleens of 6-to-8-week-old male athymic nude mice (Envigo, Indianapolis, IN, USA), as previously described [29,30]. Briefly, mice were anesthetized with 0.1 cc of ketamine (75 mg/kg I.P.) and dexmedetomidine (0.2 mg/kg I.P.), their left flank was incised and opened, and their spleen was exteriorized. Following cell injection and confirmation of a splenic subcapsular bubble, cells were allowed to dwell for 10 min—allowing for tumor cells to travel through the portal venous system and the liver to uptake them as metastases—followed by splenectomy. The abdominal contents were then returned into the abdomen and the flank was closed in two layers, followed by anesthesia reversal (atipamezole 2 mg/kg S.Q.) and analgesia (ketoprofen 4 mg/kg S.Q.). Post-operatively, mice were randomized to groups with no therapy or treatment with Gem (100 mg/kg I.P. twice weekly). A total of 40 mice (10 mice in each group) were used in this study. Metastatic tumor burden was monitored using an In Vivo Imaging System (IVIS) (Revvity, Waltham, MA, USA), which measures luciferase-expressing tumor cell bioluminescence after the administration of luciferin substrate (150 mg/kg I.P.; ChemieTek, Indianapolis, IN, USA) [31,32]. Imaging was performed starting on post-operative day 2 and continued every 3–4 days until humane endpoints were reached, at which point mice were euthanized and their liver tissue was collected for further analysis.

### 2.4. PRMT5 Pharmacologic Inhibition in the PDX Orthotopic Tumor Implantation (T.I.) Model

Pieces of human PDX tumor 366 measuring ~10 mm^3^ were surgically implanted into the pancreas of 6-to-8-week-old male athymic nude mice (Envigo) utilizing the previously described technique [26,27]. A total of 40 mice (10 mice each group) were used in this study. Briefly, mice were anesthetized with 0.1 cc of ketamine (75 mg/kg I.P.) and dexmedetomidine (0.2 mg/kg I.P.), their left flank was incised and opened, and the pancreas was exteriorized. The tumor piece was then sutured to the pancreas and the pancreas was returned into the abdomen. The flank was closed in two layers, followed by anesthesia reversal (atipamezole 2 mg/kg S.Q.) and analgesia (ketoprofen 4 mg/kg S.Q.). After 16 days, to allow for engraftment and the tumor size to reach 100–500 mm^3^, mice were randomized to groups with no therapy or treatment with Gem (10 mg/kg I.P. twice weekly) and Ptx (5 mg/kg I.P. twice weekly), a PRMT5 pharmacological inhibitor (JNJ64619178 10 mg/kg per oral administration [P.O.] daily; ChemieTek), or a combination therapy with Gem, Ptx, and the PRMT5 inhibitor. The small-molecule PRMT5 inhibitor used in this study is a selective S-adenosyl-I-methionine (SAM) mimetic/competitive inhibitor of PRMT5 [33] and was chosen due to its being investigated in a phase I clinical trial for solid organ tumors [18]. Mice were monitored with weekly MRI beginning on the day of treatment initiation and followed until humane endpoints were reached, at which point tumor tissues were weighed and collected for further analysis. MRI was carried out as described in Section 2.2.

### 2.5. Immunohistochemistry

After harvesting tumor samples at necropsy, tissues were formalin-fixed and paraffin-embedded for immunohistochemistry (IHC). IHC staining was performed by the Biorepository and Tissue Research Facility at the University of Virginia. IHC of phosphorylated gamma-H2AX (*γ*-H2AX) was employed to assess cellular responses to DNA double-strand breaks—as a sensitive marker of DNA damage and repair [34]. The primary antibody anti-*γ*-H2AX was purchased from Cell Signaling (Danvers, MA, USA). Negative control staining was performed omitting the primary antibody. The number of *γ*-H2AX-positive cells within high-power fields (HPFs) was then counted in the stained tumor sections under a microscope. For each experimental group, at least three distinct tumors were analyzed. Within each tumor, five different HPFs were selected for counting.

### 2.6. Statistical Analysis

In the orthotopic model experiments, tumor volume was averaged for each group at each MRI timepoint, and tumor weight was similarly averaged. In the metastatic model experiment, to adjust for differential tumor cell uptake within the liver, each mouse’s bioluminescence value beyond day 2 was divided by its day 2 value to produce a relative bioluminescence value. Mean relative bioluminescence values were then calculated for each group at each imaging timepoint. Continuous variables were summarized as means with standard errors of the mean (SEM) or medians with the interquartile range, as appropriate. We used GraphPad Prism to perform a Kruskal–Wallis test for nonparametric or mixed data. One-tailed Student’s *t*-tests were also used to assess differences in outcomes between two groups with a specific directional hypothesis. A *p* value < 0.05 was considered significant. GraphPad Prism software (Version 10.3.0, La Jolla, CA, USA) was used for all statistical analysis.

## 3. Results

### 3.1. PRMT5 Depletion Decreases Pancreatic Tumor Growth in Vivo

In the orthotopic pancreatic injection model (Figure 1A), the Gem-treated *PRMT5* KO tumors, compared to all other groups, demonstrated the slowest growth over the course of the treatment (Figure 2A,B). The untreated *PRMT5* KO tumors exhibited a 46% reduction in tumor volume compared to the untreated WT tumors (958.5 mm^3^ vs. 1765.5 mm^3^, *p* = 0.009) (Figure 2C). Additionally, the Gem-treated *PRMT5* KO tumors exhibited a 30% reduction in tumor volume compared to the Gem-treated WT tumors (404.9 mm^3^ vs. 579.1 mm^3^, *p* = 0.046). The tumor weights obtained at necropsy demonstrated similar trends in *PRMT5* KO efficacy (Figure 2D). The rates of liver and abdominal metastases were decreased with *PRMT5* KO or Gem treatment (Figure 2E). These findings suggest that *PRMT5* depletion significantly decreases primary tumor growth and shows synergistic inhibition with Gem of tumor growth.

### 3.2. PRMT5 Depletion Attenuates Pancreatic Tumor Metastasis in Vivo

Given the decreased metastatic rate of liver and abdominal metastases observed in the mice administered a combination treatment of PRMT5 KO with Gem in the orthotopic pancreatic injection model (Figure 2E), we employed the splenic injection model to assess the liver metastasis of mPanc96 PDAC cells (as depicted in Figure 1B). In the liver metastasis model, the combination of PRMT5 depletion with Gem treatment led to significant inhibition of liver metastasis over time (Figure 3A,B). At the final imaging timepoint, the untreated *PRMT5* KO tumors exhibited a 45% reduction in relative bioluminescence compared to the untreated *PRMT5* WT tumors (83.3 vs. 152.8, *p* = 0.018) (Figure 3C). Importantly, the Gem-treated *PRMT5* KO tumors exhibited a 52% reduction in bioluminescence compared to the Gem-treated *PRMT5* WT tumors (17.2 vs. 36.0, *p* = 0.015) (Figure 3C). Furthermore, the Gem-treated *PRMT5* KO tumors demonstrated an 88.7% decrease in relative bioluminescence compared to the untreated *PRMT5* WT tumors (17.2 vs. 152.8, *p* < 0.001) (Figure 3C). These results indicate that *PRMT5* depletion significantly decreases liver metastasis and demonstrates an additive inhibition with Gem.

### 3.3. Combination Therapy of Pharmacologic PRMT5 Inhibition with Chemotherapeutics Synergistically Inhibits PDAC Tumor In Vivo

Given that the current first- or second-line treatment for patients with stage I–IV pancreatic cancer is Gem plus Ptx, we sought to assess the effect of a PRMT5 pharmacologic inhibitor in combination with Gem plus Ptx. In the orthotopic tumor implantation model (depicted in Figure 1C), we first titrated the combination of Gem plus Ptx to find the ideal dose that would allow us to assess the combination plus the PRMT5 inhibitor (PRMT5i) (Appendix A). Given that 10 mg/kg Gem plus 5 mg/kg Ptx resulted in a significant inhibition but still allowed tumor growth, it was determined that it would be the best dose to use to test the combination treatment (Gem, Ptx, and PRMT5i). This triple therapy was most effective at inhibiting tumor progression (Figure 4A). The tumors treated with PRMT5i alone, as compared to the control-treated tumors, demonstrated a 20% decrease in volume at the end of the experiment (2177 mm^3^ vs. 2730 mm^3^, *p* = 0.16; Figure 4B). Similarly, the tumors treated with the combination therapy revealed the smallest final volume, a 65% reduction compared to the control (966 mm^3^ vs. 2730 mm^3^, *p* = 0.001). The combination therapy-treated tumors, compared to the Gem and Ptx-treated tumors, showed an 18% decrease in tumor volume (966 mm^3^ vs. 1184 mm^3^, *p* = 0.13; Figure 4B). The decrease in tumor volume, which was not statistically significant, may be due to some inherent errors in the MRI volume assessment of irregularly shaped tumors. In addition, a less effective dose of Gem plus Ptx would likely show significance in comparison to the combination therapy and may be tested in the future. The comparison of tumors at necropsy corroborated these findings, with the combination therapy-treated tumors demonstrating a 26% lower weight than the Gem and Ptx-treated tumors (1.09 g vs. 1.47 g, *p* = 0.037; Figure 4C). Importantly, the addition of PRMT5i to the control or Gem/Ptx treatment had a limited impact on toxicity, reflected by the similar mouse weights among the treated groups over the course of the treatment (Figure 4D). These findings suggest that the combination therapy of pharmacologic PRMT5 inhibition with chemotherapeutics inhibits PDAC tumors in vivo.

### 3.4. PRMT5 Depletion Increases DNA Damage in Pancreatic Tumors

We then aimed to understand the mechanism behind the combination therapy and assess the therapeutic value of this combination in vivo. Counting the *γ*-H2AX-positive cells per HPF allows for a quantitative assessment of the level of DNA damage within the tumor [35]. A higher number of *γ*-H2AX-positive cells per HPF typically indicates increased DNA damage, potentially linked to treatment response. IHC of the tumors obtained at necropsy demonstrated a significant increase in the expression of *γ*-H2AX, a marker of DNA damage in the form of DNA double-strand breaks, in the treatment of *PRMT5* KO cells with Gem (Figure 5). These results suggest that PRMT5 depletion significantly induces DNA damage in combination therapy with chemotherapeutics.

### 3.5. Summary of Effects of Combination Therapy of PRMT5 Inhibition with Chemotherapeutics

Figure 6 provides a schematic representation of the synergistic effects of PRMT5 inhibition with chemotherapy, highlighting a potential combination therapy for PDAC. PRMT5 activates RPA2-mediated DNA repair pathways in chemo-treated cells [8]. The depletion or inhibition of PRMT5 synergistically induces apoptosis through blocking the RPA-mediated DNA repair pathway in response to chemotherapy.

## 4. Discussion

Our main objective in this study was to evaluate the efficacy of PRMT5 inhibition in combination with Gem-based chemotherapy using clinically relevant orthotopic and metastatic PDX mouse models of pancreatic cancer. These mouse models closely resemble the human tumor environment, which is crucial for understanding cancer progression and metastasis. The data generated in these models may more accurately predict drug efficacy in clinical trials. In three different preclinical models of PDAC, PRMT5 depletion or inhibition inhibited primary PDAC growth and metastasis in combination with standard-of-care chemotherapy, suggesting that PRMT5 is a potential target for combination therapy. The addition of a PRMT5 inhibitor to Gem plus Ptx therapy was tolerated. The combination of the PRMT5 inhibitor plus Gem was significantly better than Gem alone or the PRMT5 inhibitor alone, and the combination of the PRMT5 inhibitor plus Gem plus Ptx was slightly better than just Gem plus Ptx. Our prior work employing a traditional subcutaneous model demonstrated the substantial potential of combining PRMT5 inhibition and Gem for the treatment of PDAC [8]. This difference is likely due to orthotopic and metastatic models being more complex than traditional subcutaneous models, requiring larger group sizes to reliably assess drug effects. However, our orthotopic and metastatic PDX models offer a more clinically relevant platform for drug development and treatment evaluation.

Preclinical models remain a critical step in translating scientific discovery into improvements in patient care. Most novel therapies fail to demonstrate efficacy within human clinical trials [36], a failure which many ascribe to poorly designed and biologically irrelevant preclinical models [37,38]. For example, therapies tested in genetically engineered mouse models of PDAC are often initiated immediately after tumor development—a practice which fails to capture the advanced or metastatic disease with which 80% of patients with PDAC present [39]. Similarly, PDX models are frequently performed in heterotopic positions, such as the subcutaneous space of the flank, which inadequately reflects the tumor microenvironment of the pancreas and rarely develops metastases [24,37]. While no model is perfect, the mindful execution of clinically and biologically relevant models, such as those presented in this study, can help identify patients and disease settings that may benefit from therapy.

The current findings warrant continued evaluation of PRMT5 inhibitors in combination with Gem-based chemotherapy. Indeed, some clinical trials have already been initiated using RAS (ON) or CDK4/6 inhibition in combination with PRMT5 inhibition [40,41]. Future work should focus on newer pharmacologic inhibitors, as well as additional targets, to improve the efficacy of PRMT5 inhibition. JNJ64619178 primarily targets the PRMT5/methylosome protein 50 (MEP50) enzymatic complex, which has high-affinity binding to both the S-adenosylmethionine (SAM) and protein substrate-binding pockets, and inhibits its function [20,42,43]. Studies have shown that at high concentrations, JNJ64619178 inhibits PRMT5/MEP50 by over 80% while causing minimal inhibition (<15%) of other closely related arginine methyltransferases like PRMT1 and PRMT7, suggesting that JNJ64619178 is highly selective for PRMT5, with minimal off-target effects on other methyltransferases. On the other hand, MRTX1719 selectively targets the PRMT5/metabolite methylthioadenosine (MTA) complex, which is elevated in cancers with the deletion of methylthioadenosine phosphorylase (MTAP). MTAP deletion leads to an accumulation of the metabolite methylthioadenosine (MTA), which can act as a competitive inhibitor of SAM [21,44]. Overall, JNJ64619178 is a highly selective PRMT5 inhibitor, demonstrating minimal inhibition of other methyltransferases. Furthermore, these results suggest that the pharmacologic inhibition of PRMT5 alone for the treatment of solid organ tumors, as is currently being investigated in several ongoing clinical trials [18,19,20,21,22,23], may be of limited utility. Thus, combination cancer therapy involving two or more treatments, like a PRMT5 inhibitor combined with chemotherapy as used in this study, targets PDAC cells more effectively and improves treatment outcomes. This combination approach aims to reduce the risk of cancer cells developing resistance, allow for optimal dosages of each treatment, and potentially achieve synergistic effects where the combined treatments work together more powerfully than individually [45]. Combination therapy, using various mechanisms to target cancer cells, reduces the likelihood that cancer cells will develop resistance to either specific treatment [46]. Thus, combining treatments can lead to more effective cancer cell destruction and better control of tumor growth. 

In addition to PRMT5 as a therapeutic target, there are other possible options. PRMT5 inhibition is hypothesized to synergize with Gem through the exhaustion of the RPA pool needed to protect ssDNA at stalled replication forks during Gem-induced replication stress, resulting in replication catastrophe [8,47]. However, given that higher RPA levels are associated with increased cancer therapeutic resistance [48,49], it is possible that more effective targeting of the RPA pathway is needed [50]. RPA is essential for nearly all aspects of DNA metabolism and, in particular, homologous recombination (HR)-mediated DNA repair [51]. PRMT5 activates RPA2-mediated DNA repair pathways in response to chemotherapy [8]. Collectively, these findings suggest that RPA plays a central role in the combination therapy of PRMT5 inhibition with chemotherapeutics. The inhibition of RPA2 could be essential in achieving PDAC cell replication catastrophe and death. An RPA2 inhibitor may be an alternative drug for PRMT5 inhibition and warrants testing in preclinical models as part of a combination therapy with chemotherapeutics. Targeting RPA in combination with chemotherapeutics can lead to synthetic lethality, where cancer cells become more vulnerable to the combination therapy.

However, to identify a favorable target in the RPA pathway, the mechanism by which PRMT5 depletes RPA requires further investigation. One hypothesis for the role of PRMT5 in RPA modulation involves the MRE11 enzyme, which is rapidly recruited to DNA double-strand breaks (DSBs) to initiate repair [52]. The MRE11 protein contains a glycine–arginine-rich (GAR) motif, which, when methylated by protein arginine methyltransferases (PRMTs), is critical for DNA binding [53]. Genetic substitution of the arginine residues on GARs with lysine residues—thus preventing PRMT methylation—causes significant decreases in RPA–ssDNA complex formation and impaired HR after DNA damage [54]. Similarly, another postulated mechanism for the effect of PRMT5 on RPA and HR is through the arginine methylation of RUVBL1 (Pontin/Tip49), a cofactor of the TIP60 complex that mobilizes 53BP1 from DNA breaks and drives HR [55]. The silencing of PRMT5 and, separately, the de-methylation of RUVBL1 significantly reduce RPA foci and the activity of downstream effectors of HR—demonstrating the importance of the PRMT5 methylation of RUVBL1 in HR. Given the diversity of targets that PRMT5 methylates, it is plausible that multiple mechanisms may drive RPA exhaustion during PRMT5 inhibition. Accordingly, future work should seek to elucidate the mechanisms by which PRMT5 regulates RPA and identify which steps in this pathway are most vulnerable as combinatorial targets with PRMT5 inhibition.

## 5. Conclusions

In summary, this study revealed that the genetic depletion or pharmacologic inhibition of PRMT5 in clinically relevant models of PDAC is a promising strategy in combination with standard-of-care Gem-based therapy. Furthermore, the identification of RPA2 as a possible driver of the response to PRMT5 inhibition will require further research in pancreatic cancer. These results have important implications for ongoing clinical trials studying PRMT5 inhibitors in solid tumor growth and metastasis and highlight the importance of rigorous translational investigations of promising novel therapies.

## Figures and Tables

**Figure 1 biomolecules-15-00948-f001:**
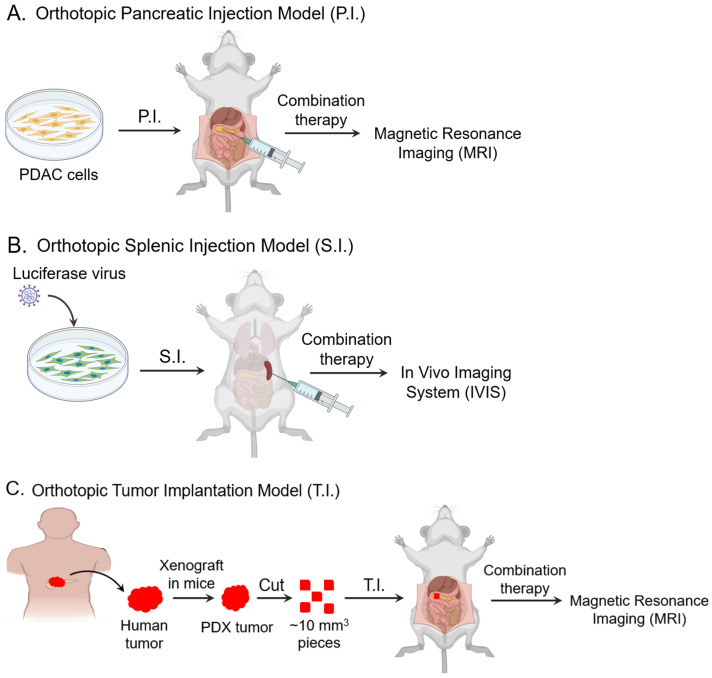
Workflows of the orthotopic mouse models used in this study. (**A**) In the pancreatic injection (P.I.) model, PDAC cells were orthotopically injected into the pancreas of nude mice, and tumor growth was monitored by MRI. (**B**) In the splenic injection (S.I.) model, luciferase-transduced PDAC cells were injected into the spleens of nude mice, and metastatic tumor burden was monitored using IVIS bioluminescence. (**C**) In the human PDAC tumor implantation (T.I.) model, ~10 mm^3^ pieces of human PDX tumor were surgically implanted into the pancreas of nude mice, and tumor growth was monitored by MRI. Cell culture and mouse templates were taken from the Biorender.com.

**Figure 2 biomolecules-15-00948-f002:**
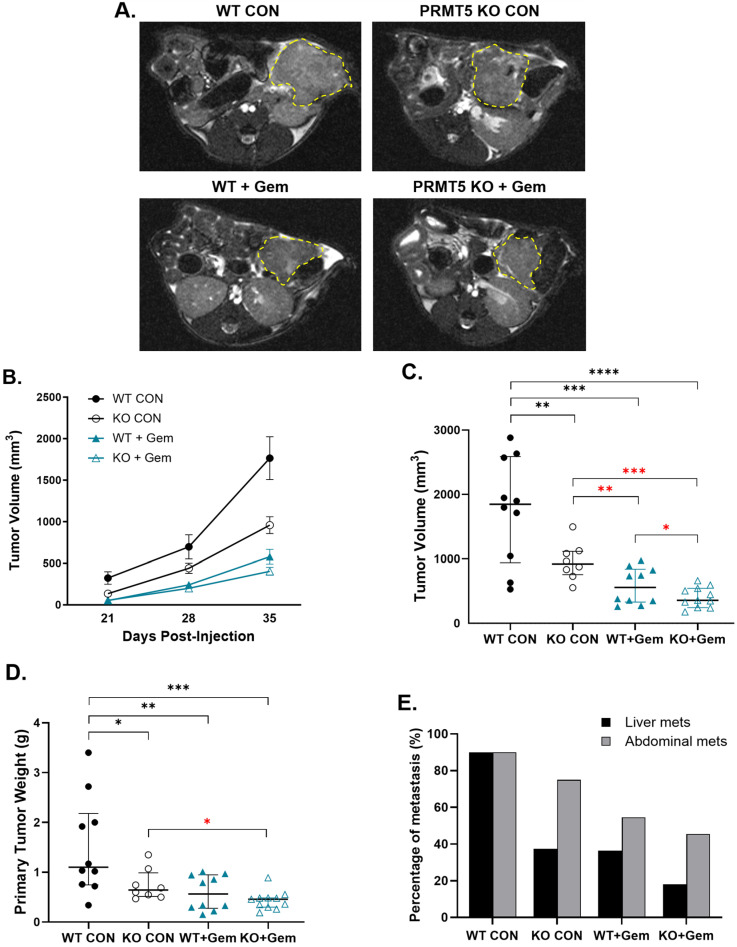
Effect of *PRMT5* depletion on pancreatic tumor growth and metastasis in an orthotopic P.I. model. In total, 1 × 10^6^ WT or *PRMT5* KO mPanc96 cells were injected into the pancreas of nude mice and subsequently treated with Gem. Tumor growth was monitored by MRI. Representative axial MRI images at 35 days post-injection of control (CON) and Gem-treated WT and *PRMT5* KO tumors are shown, with the tumor cross-sectional area outlined (**A**). Average (mean +/− SEM) tumor volume as measured by MRI over time (**B**). The median and interquartile range for tumor volume (**C**), tumor weight (**D**), and metastatic tumor percentage (**E**) were assessed at the end of the experiment (35 days post-injection). *n* = 10 mice per group; Kruskal–Wallis test was used to analyze nonparametric or mixed data, * *p* < 0.05, ** *p* < 0.01, *** *p* < 0.001, **** *p* < 0.0001; Student’s *t*-tests were used to assess the differences between two groups with a specific directional hypothesis (red asterisk), *
*p* < 0.05, **
*p* < 0.01, ***
*p* < 0.001.

**Figure 3 biomolecules-15-00948-f003:**
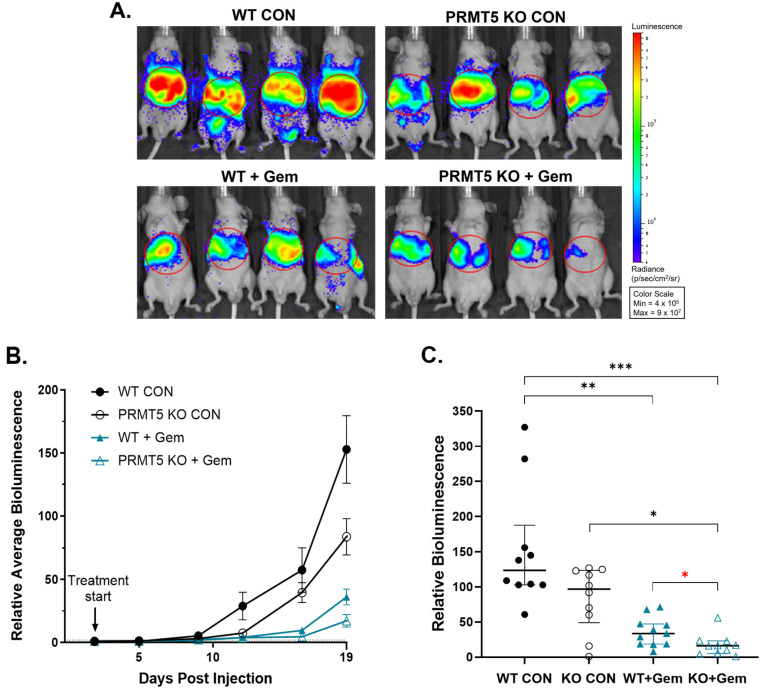
Effect of *PRMT5* depletion on splenic tumor metastasis in an S.I. model of liver metastasis. In total, 2 × 10^6^ WT or *PRMT5* KO mPanc96 cells with luciferase virus infection were injected into the pancreas and subsequently treated with Gem. Tumor growth was monitored by bioluminescence. Representative bioluminescence images at the experiment endpoint (19 days post-injection) are shown, where red indicates maximum bioluminescence, blue indicates minimal bioluminescence, and the red line indicates the region of interest used in the analysis (**A**). Average (mean +/− SEM) bioluminescence, relative to post-injection day 2, over time (**B**). Relative bioluminescence (median with interquartile range) at end of experiment, 19 days post-injection (**C**). *n* = 10 mice per group; Kruskal–Wallis test was used to analyze nonparametric or mixed data, * *p* < 0.05, ** *p* < 0.01, *** *p* < 0.001; Student’s *t*-tests were used to assess the differences between two groups with a specific directional hypothesis (red asterisk), *
*p* < 0.05.

**Figure 4 biomolecules-15-00948-f004:**
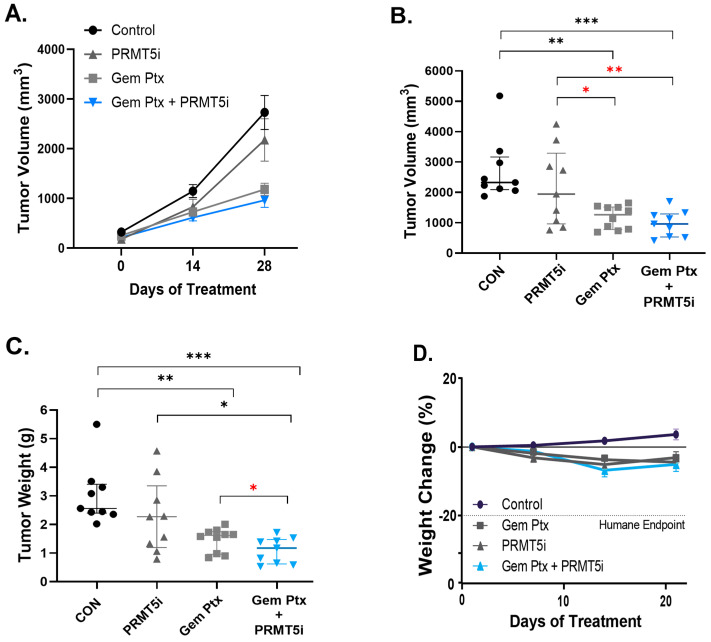
Effect of PRMT5 pharmacologic inhibition on pancreatic tumor growth in orthotopic T.I. model. Pieces of human PDX tumor measuring ~10 mm^3^ were surgically implanted into the pancreas of nude mice, and tumor growth was monitored by MRI. The mice were treated with a PRMT5 inhibitor [PRMT5i] or/and Gem and Ptx as indicated, and the average (mean +/− SEM) bioluminescence, relative to post-injection day 2, was measured over time (**A**). The median and interquartile range for tumor volume (**B**) and tumor weight (**C**) were assessed at the end of the experiment (28 days of treatment). The mouse weight change over the course of the treatment in each treatment group (**D**). *n* = 10 mice per group; Kruskal–Wallis test was used to analyze nonparametric or mixed data, * *p* < 0.05, ** *p* < 0.01, *** *p* < 0.001; Student’s *t*-tests were used to assess the differences between two groups with a specific directional hypothesis (red asterisk), *
*p* < 0.05, **
*p* < 0.01.

**Figure 5 biomolecules-15-00948-f005:**
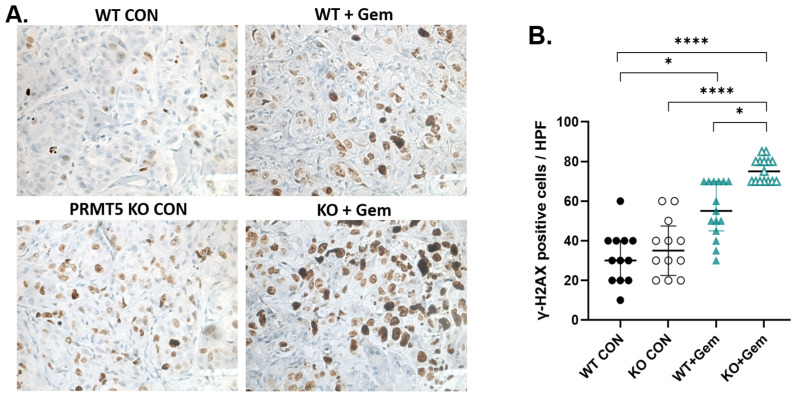
PRMT5 depletion results in increased DNA damage in PDAC tumors. IHC staining of *γ*-H2AX, a marker of DNA damage, was performed using tumor sections from the orthotopic P.I. model using mPanc96. (**A**) Representative IHC staining for *γ*-H2AX in tumor sections (magnification 40×). (**B**) *γ*-H2AX-positive cells in high-power field (HPF) were counted in the stained tumor sections generated from orthotopic pancreatic injection. *n* = 15 (3 tumors per group; 5 HPF counts in each of the 3 tumors); Kruskal–Wallis test was used, * *p* < 0.05, **** *p* < 0.0001.

**Figure 6 biomolecules-15-00948-f006:**
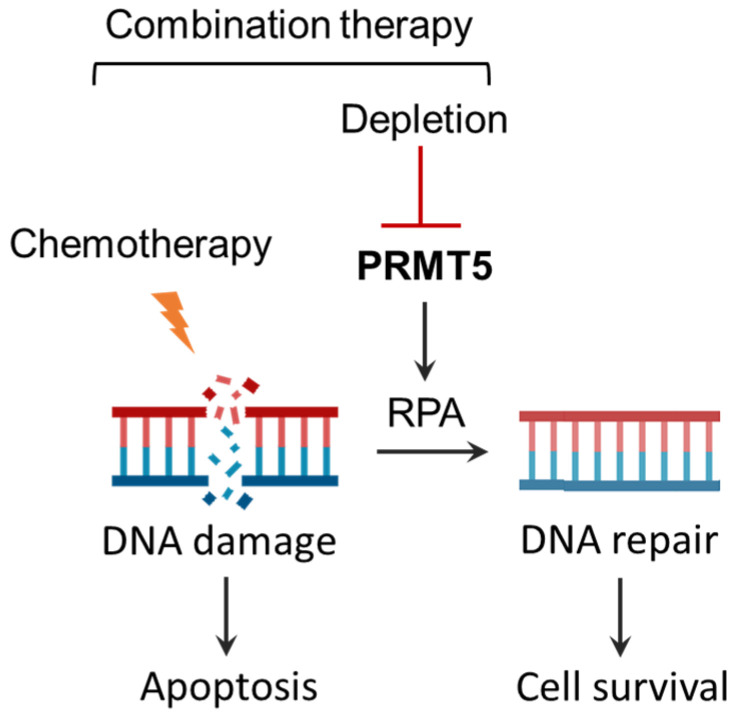
Schematic summarizing the synergistic effects of combination therapy.

## Data Availability

All the authors are affiliated with the University of Virginia School of Medicine, where the research was conducted. The data can be found via the institute’s network, and all authors had full access throughout the study. The original contributions presented in the study are included in the article; further inquiries can be directed to the corresponding author.

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
