# Peer review of "PRMT5 Identified as a Viable Target for Combination Therapy in Preclinical Models of Pancreatic Cancer"

_biomolecules, 2025, doi:10.3390/biom15070948_

Round 1

Reviewer 1 Report

Comments and Suggestions for Authors

This manuscript by Wei et al. presents a compelling preclinical investigation into the therapeutic potential of PRMT5 inhibition in pancreatic ductal adenocarcinoma (PDAC). The use of orthotopic and metastatic patient-derived xenograft (PDX) models adds significant translational relevance. The findings are timely and well-supported by the data, particularly the demonstration of enhanced efficacy of gemcitabine (Gem) and paclitaxel (Ptx) when combined with PRMT5 depletion or inhibition. The mechanistic insight into DNA damage via γ-H2AX staining is a notable strength. However, several areas require clarification or expansion to strengthen the manuscript’s impact and translational potential.

Major Comments
  1. Both genetic knockout and pharmacologic inhibition of PRMT5 were used in this study, but rescue experiments are lacking, which would substantially strengthen the specificity claims.
  2. The authors should also comment on the selectivity profile of JNJ64619178—particularly its off-target effects or impact on other methyltransferases—and explain why this inhibitor was chosen over more clinically advanced alternatives like MRTX1719.
  3. Please provide power calculations or effect size estimates for the main endpoints (e.g., tumor volume reduction), especially for Figure 4, where the difference between PRMT5i+Gem+Ptx and Gem+Ptx groups did not reach significance (p=0.13 for tumor volume).
  4. The use of one-tailed t-tests should be justified or replaced with two-tailed tests unless a strong directional hypothesis was pre-specified.
Minor Comments
  1. Improve figure legends for clarity—some are overly brief or lack sufficient details

Author Response

We greatly thank the reviewer for their perspectives and detailed examination of our manuscript. We have carefully revised the manuscript to better relate our findings to the literature (i.e., rescue experiments, PRMT5 inhibitors) in the discussion section. We have also provided a more detailed explanation of the power calculations or effect size estimates for the main endpoints in the main text. Our point-wise responses to the reviewer’s comments are detailed in the attached file.

Note: Our response to the reviewer comments is in blue, and the edits made to the manuscript are highlighted in yellow.

Reviewer 2 Report

Comments and Suggestions for Authors

The manuscript is devoted to the study of a topical and clinically significant problem—the development of new therapeutic strategies for the treatment of aggressive forms of pancreatic cancer, namely pancreatic ductal adenocarcinoma (PDAC). The authors consider PRMT5 (protein arginine methyltransferase 5) as a potential target for targeted therapy in combination with traditional chemotherapeutic agents such as gemcitabine and paclitaxel.

The work was performed at a high level using clinically relevant orthotopic and metastatic mouse models, including a PDX model, which enhances its significance for the development of personalized medicine approaches. The authors claim that PRMT5 depletion, both through genetic knockout and pharmacological inhibition, leads to a reduction in primary tumor volume, a decrease in metastases, and an increase in DNA damage during chemotherapy treatment. Based on the results presented, the authors emphasize the promise of PRMT5 as a therapeutic target in a combined approach to PDAC therapy.

Nevertheless, despite the well-thought-out experimental design, there are some points in the paper that raise questions, primarily concerning statistical data processing and interpretation of results. Below are the main comments and questions for the authors that will help make the study more accurate and convincing:

Comments on the «Materials and Methods» section

Sections 2.1.-2.4. Please specify the exact number of animals in each experimental group for all models used.

2.5. Immunohistochemistry

The IHC protocol is incomplete. It is recommended to describe in detail how the quantitative analysis of the data was performed.

2.6. Statistical Analysis

There are doubts about the correctness of the chosen statistical approach.

1) Was the data checked for normal distribution? If so, what method was used? In the case of abnormal data distribution, it is incorrect to use the mean, SEM, and SD for statistical analysis and presentation of results; it is necessary to use the median and interquartile range.

2) The use of the t-test is incorrect for comparing three or more groups. In such cases, it is recommended to use tests that consider multiple comparisons.

Comments on the «Results» section

- Figure 2 (E). Line 178: metastatic tumor percentage.

There is no data scatter on the graph. It is not entirely clear how metastasis was assessed and what was taken as 100%. The data scatter should be presented and the method of assessing metastasis should be clarified.

- Line 197: pancreas.

There is probably a mistake here — in the model of splenic cell injection, it should apparently refer to the spleen, not the pancreas.

- Lines 213-215: Combination-treated tumors, compared to Gem and Ptx-treated tumors, showed an 18% decrease in tumor volume (966 mm3 vs. 1184 mm3, p=0.13; Figure 4B).

When comparing the «Gem,Ptx» and «Gem,Ptx+PRMT5i» groups, there is no statistical significance (Figure 4B). However, in the assessment of tumor weight (Figure 4C), the significance is at the threshold of reliability. A statistical analysis should be performed using correct methods for comparing three or more groups and considering the multiplicity of comparisons.

- Line 229: 3.4. PRMT5 depletion increases DNA damage in pancreatic tumors.

It is recommended to clarify how the quantitative assessment of the data was performed. It is recommended to provide representative images of stained sections.

- Line 234: PRMT5 depletion or inhibition.

The data presented for γ-H2AX refer only to PRMT5 KO, but not to PRMT5 inhibition. Either add the relevant data or clarify the wording so as not to mislead the reader.

- Line 234: synergistically

There is no assessment of synergy using appropriate methods (no calculations, no data showing that a synergistic effect is occurring). This term should be used with caution, or the sentence should be reworded.

- Line 235: induces DNA damage

Graph 5 does not indicate whether there are significant differences between the «WT+Gem» and «KO+Gem» groups and the «WT» and «WT+Gem» groups. This is important for assessing the accuracy of the conclusions presented regarding the contribution of PRMT5 depletion and Gem to DNA damage.

Due to the need to revise the statistical processing of the data, the «Discussion» section may also need to be revised in accordance with the results obtained.

Author Response

We greatly thank the reviewer for their detailed examination of our manuscript, positive feedback, and helpful comments. We believe that we have addressed all the reviewer’s concerns in the attached file and have revised the manuscript accordingly. We greatly appreciate the reviewer for their comments on statistical analysis.

Note: Our response to the reviewer comments is in blue, and the edits made to the manuscript are highlighted in yellow.

Round 2

Reviewer 1 Report

Comments and Suggestions for Authors

Thank you for your thorough and thoughtful revision. The authors have addressed all the previous comments and concerns, and the revised manuscript has improved in clarity, scientific rigor, and overall presentation.

The additional data and discussions enhance the strength of the findings, and the responses to reviewer feedback were comprehensive and satisfactory. I commend the authors for their efforts in refining the manuscript.

I have no further major concerns and am pleased to recommend acceptance of the manuscript in its current form. I wish you success with the publication and future research in this area.

Reviewer 2 Report

Comments and Suggestions for Authors

Dear authors,

I have carefully reviewed the revised version of the article and am pleased to note that my comments have been taken into account. The results have been substantially revised, correct conclusions have been drawn, and as a result, the work has acquired a more complete and presentable appearance.